

# `treedata.table`: a wrapper for `data.table` that enables fast manipulation of large phylogenetic trees matched to data

Cristian Román Palacios[1,2], April Wright[3] and Josef Uyeda[4]

[1] Department of Atmospheric and Oceanic Sciences; Department of Earth, Planetary, and Space Sciences, Institute of the Environment and Sustainability; Center for Diverse Leadership in Science, University of California, Los Angeles, Los Angeles, CA, United States of America
[2] School of Information, University of Arizona, Tucson, AZ, United States of America
[3] Biology Department, Southeastern Louisiana University, Hammond, LA, United States of America
[4] Department of Biological Sciences, Virginia Polytechnic Institute and State University (Virginia Tech), Blacksburg, VA, United States of America

## ABSTRACT

The number of terminals in phylogenetic trees has significantly increased over the last decade. This trend reflects recent advances in next-generation sequencing, accessibility of public data repositories, and the increased use of phylogenies in many fields. Despite R being central to the analysis of phylogenetic data, manipulation of phylogenetic comparative datasets remains slow, complex, and poorly reproducible. Here, we describe the first R package extending the functionality and syntax of `data.table` to explicitly deal with phylogenetic comparative datasets. `treedata.table` significantly increases speed and reproducibility during the data manipulation steps involved in the phylogenetic comparative workflow in R. The latest release of `treedata.table` is currently available through CRAN (https://cran.r-project.org/web/packages/treedata.table/). Additional documentation can be accessed through rOpenSci (https://ropensci.github.io/treedata.table/).

## INTRODUCTION

The number and size of published phylogenetic trees have exponentially increased over the years (Fig. 1; *Smith et al., 2011*; *Fitzjohn et al., 2014*; *Smith & Brown, 2018*). Ongoing biodiversity sequencing efforts have triggered the development of phylogenetic computational methods able to deal with datasets involving hundreds of thousands of taxa (*McMahon et al., 2015*). For instance, the early development of MAFFT (*Katoh, 2002*) significantly decreased computational times required to perform sequence alignment on molecular datasets with thousands of species. Similarly, RAxML (*Stamatakis, 2006*), PATHd8 (*Tamura et al., 2012*), and TreePL (*Smith & O'Meara, 2012*) greatly reduced computational times during the inference and absolute dating of phylogenetic trees including thousands of species. Given the unprecedented pace at which phylogenetic data is accumulating

Corresponding author
Cristian Román Palacios,
cromanpa@g.ucla.edu,
cromanpa94@arizona.edu

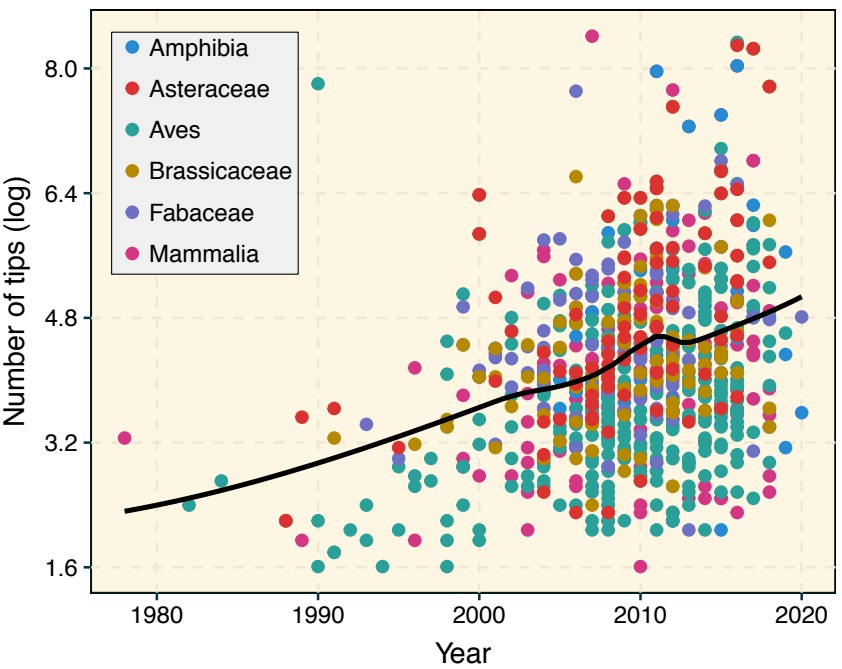

**Figure 1** **Temporal change in phylogenetic tree sizes between 1978 and 2020 based on 927 publications for different animal and plant groups.** We used a LOESS smoothing to depict the temporal trend in tree size over time. Data was retrieved from the Open Tree of Life (*Redelings et al., 2019*) using the rotl R package (*Michonneau, Brown & Winter, 2016*). A linear regression that accounted for lineage identity indicated the significant increase in tree size over time ($R^2 = 0.2077$, $p < 0.001$).

(*Piel et al., 2000*; *Redelings & Holder, 2017*), updating the current comparative phylogenetic workflow to cope with the increasing size of phylogenetic trees is now more critical than ever. Attention should be paid to the development of faster, computationally efficient, and more user-friendly implementations in R that further increase reproducibility. The R language (*R Core Team, 2013*) is now central to research utilizing phylogenetic comparative methods, and many essential packages and educational materials are made available using this language (*Harmon, 2019*). The latest release of treedata.table is available through CRAN (https://cran.r-project.org/web/packages/treedata.table/). More information about the treedata.table R package can be found in rOpenSci (https://ropensci.github.io/treedata.table/).

## A SHORT DESCRIPTION OF DATA.TABLE

treedata.table heavily relies on data.table, an R package that enables high-performance extended functionality for data tables (*Dowle & Srinivasan, 2019*). data.table is not only faster than other packages implemented in R, but also is significantly more efficient than tools in other languages such as Python and Julia (*db-benchmark project, 2021*). In addition to speed, data.table has a syntactic structure that is clear and simple to follow. Only three elements are basic to data.table's primary function: DT[i, j, by]. First, the i section is used to specify the rows to be considered in filtering

or subsetting operations. Second, the `j` section indicates the changes happening in the columns (*e.g.*, adding new ones, changing existing ones). Third, the by section is used to perform operations based on grouping variables. A brief but more exhaustive introduction to `data.table` can be found in the `data.table`'s vignette and wiki.

Data manipulation can be performed through numerous approaches in R. Each of these alternatives have their own particular advantages. For instance, packages in the `tidyverse` (*e.g.*, `dplyr`) are in general designed to increase readability and flexibility during data munging steps (*Wickham et al., 2019a*; *Wickham et al., 2019b*). Data wrangling in `base R` is largely standard and in general more stable over time than other approaches. Here, we focus on extending the functionality of `data.table`, a package that is generally faster and more concise than other approaches, for dealing with phylogenetic comparative datasets.

### The `treedata.table` workflow

`treedata.table` is a wrapper for `data.table` designed for phylogenetic analyses that matches a phylogeny to a `data.table` (Table 1). After an initial tree/data matching step, `treedata.table` continuously preserves the tree/data matching across `data.table` operations. `treedata.table` also allows users to run functions from other phylogenetic packages on the processed `treedata.table` objects. Below, we briefly explain the general workflow under `treedata.table`.

(1) **Tree and character matrix matching:** Using the `treedata.table` package begins with creating a `treedata.table` object. as.`treedata.table` function matches the `tip.labels` of the phylogeny to a column of names in the `data.frame`.

(2) **`treedata.table` operations:** two main functions allow users to make changes to `treedata.table` objects. Changes are reciprocal between trees and data.

(A) **Explicitly dropping taxa:** Taxa in `treedata.table` objects can be dropped using the `droptreedata.table` function. Dropped taxa results are removed from the character matrix and trees.

(B) **Data operations:** The most powerful functionality of `treedata.table` is related to functions calling `data.table`. The [ function, taking the same arguments as the analog function in `data.table`, can be used to subset rows, select, and/or compute statistics on columns in the character matrix of the `treedata.table` object (`DT[i, j, by]`). Operations changing the number of rows in the character matrix will also affect the corresponding taxa in the tree.

(3) **Data extraction from `treedata.table` objects:** Users can independently extract trees and character matrices from `treedata.table` objects using the `pulltreedata.table` function. The $ operator is also a valid alternative to `pulltreedata.table`. Two additional functions ([[ and `extractVectors`) can be used to extract named vectors from `treedata.table` objects. These operations streamline formatting of data into the various different input requirements of R functions from other phylogenetics packages.

(4) **Using external functions in `treedata.table` objects:** the `tdt` function enables users to easily run external functions on `treedata.table` objects directly. Specifically, `tdt` passes data and tree attributes from a given `treedata.table` object as arguments to functions implemented in other packages.

**Table 1 Brief descriptions of the functions implemented in `treedata.table`.** We list functions under eight different categories and provide a brief outline of their main uses.

| Category | Function | Description |
|---|---|---|
| treedata.table object creation | as.treedata.table | Initial step of the workflow in treedata.table. Matches a character matrix (of class data.frame) to a single (of class phylo) or multiple trees (class multiPhylo) |
| Drop taxa from treedata.table objects | droptreedata.table | Drops taxa from a treedata.table object |
| Data manipulation | [ | Performs data.table operations on an object of class treedata.table |
| Data extraction | [[ | Extracts a named vector from an object of class treedata.table |
| | extractVector | Returns a named vector from a treedata.table object |
| | pulltreedata.table | Returns a character matrix or tree(s) from a treedata.table object |
| Run functions from other packages | tdt | Runs a function on a treedata.table object |
| Detect character type | detectCharacterType | Detects whether a character is continuous or discrete |
| | detectAllCharacters | Applies detectCharacterType over an entire character matrix |
| | filterMatrix | Filters a matrix, returning either all continuous or all discrete characters |
| Examine treedata.table objects | summary | Summarizes treedata.table objects by presenting the number of discrete and continious characters, missing values, and general changes to the original treedata.table object |
| | print | Print method treedata.table objects |
| | head, tail | Returns the first or last part of an treedata.table object |
| Inspect column/row names | hasNames | Row and column name check |
| | forceNames | Force names for rows, columns or both |

(5) **Additional functions**: treedata.tree includes additional functions to detect and filter character matrices by character types (continuous or discrete; `detectCharacterType`, `detectAllCharacters`, and `filterMatrix`). Other functions can be used to examine (`head`, `tail`, `print`) and describe (`summary`) objects of class `treedata.table`. Finally, two additional functions can be used to inspect and force column and row names in character matrices (`hasNames`, `forceNames`).

## Using `treedata.table`

This brief step-by-step tutorial is based on `treeplyr`'s *Anolis* example data, including 100 tips and 11 characters (see also Appendices S1–S2):

```
library(treedata.table)
data(anolis)
```

To use all the functionalities in `treedata.table`, we first construct a `treedata.table` object using the `as.treedata.table` function, which performs an exact name match between the tip labels of the tree and the column in the dataset with the most matches.

```
td <- as.treedata.table(tree = anolis$phy, data = anolis$dat)
```

The resulting object can be inspected using the `summary()`, `head()`, `tail()`, and `print()` functions. For instance, we can see a description of the `treedata.tree` object using the `summary()` function:

```
summary(td)
```

Next, we can perform data manipulation steps on the resulting `treedata.table` object. For instance, we can extract the SVL column (snout-vent length) using the $ function and [ operator, as follows:

```
td$dat[,'SVL']
```

A named vector of the same trait (SVL) can also be extracted using `td[["SVL"]]` or `extractVector(td, 'SVL')`. However, `extractVector` further supports extraction of multiple traits. For instance, the following code will extract two named vectors: one for SVL and another for ecomorph.

```
extractVector(td, 'SVL','ecomorph')
```

The real power in `treedata.table` is in co-indexing the tree and table based on functions from `data.table`. We can use `data.table` syntax to subset the `treedata.table` object and include only the first representative from each ecomorph in the *Anolis* dataset.

```
td[, head(.SD, 1), by = "ecomorph"]
```

We can also subset the *Anolis* dataset to include a single species per ecomorph and island:

```
td[, head(.SD, 1), by = .(ecomorph, island)]
```

Furthermore, we can create a new variable summarizing SVL+hostility for only Cuban anoles:

```
td[island == "Cuba", .(Index = SVL + hostility)]
```

While the options for data manipulations are infinite, the matching between the tree and data attributes is always constant. Finally, users can pass data and trees in `treedata.table` objects as arguments to functions in other packages. For instance, below we use the `tdt` function in `treedata.table` to fit a continuous model of trait evolution for SVL in `geiger` (*Harmon et al., 2008*; *Pennell et al., 2014*):

```
tdt(td, fitContinuous(phy, extractVector(td, 'SVL'), model="BM"))
```

All the functions explained above can handle multiple trees. For instance, below we fit the same model of continuous trait evolution on SVL based on a `multiPhylo` tree for the *Anolis* dataset:

```
trees <- list(anolis$phy,anolis$phy)
class(trees) <- ``multiPhylo''

td <- as.treedata.table(tree=trees, data=anolis$dat)

tdt(td, fitContinuous(phy, extractVector(td, `SVL'), model="BM"))
```

The introductory vignette to `treedata.table` (https://ropensci.github.io/treedata.table/articles/AA_treedata.table_intro_english.html, https://ropensci.github.io/treedata.table/articles/AB_treedata.table_intro_spanish.html) contains further information on the functions outlined above and in Table 1.

**Table 2  Functions in different R packages (including `treedata.table`) with similar functions on matched tree/data objects.**

| Package | Function | Tree/data-matched object manipulation | Reference |
|---|---|---|---|
| `treedata.table` | `as.treedata.table` | `data.table` syntax | This study |
| `geiger` | `treedata` | Not supported | *Harmon et al. (2008)*, *Pennell et al. (2014)* |
| `tidytree` | `treedata` | dplyr verbs after using `tibble::as_tibble()` | *Yu (2021)* |
| `treeplyr` | `make.treedata` | dplyr verbs | *Uyeda & Harmon (2020)* |

## COMPUTATIONAL PERFORMANCE

### Alternatives to `treedata.table`

Keeping trees and data objects separated in the R environment is a standard practice. Changes to trees and data are typically performed independently using a combination of functions implemented in ape (*Paradis & Schliep, 2018*), base (*R Core Team, 2013*), `data.table` (*Dowle & Srinivasan, 2019*), or in the `tidyverse` (*Wickham et al., 2019a*; *Wickham et al., 2019b*). However, to our knowledge, `treeplyr` (*Uyeda & Harmon, 2020*) and `tidytree` (*Yu, 2021*), both based on dplyr (*Wickham et al., 2019a*; *Wickham et al., 2019b*), are to our knowledge, the only packages that are able to perform simultaneous operations on combined tree/data objects in R (Table 2). We note that while `data.table`, `treeplyr`, and dplyr share similar functionalities, their philosophy and syntax are strikingly different. Differences between these packages ultimately relate to "source" package they rely on (*i.e.*, `data.table` or dplyr). For instance, although `data.table` uses shorter syntax relative to dplyr, the pipe operator and the use of verbs in dplyr makes this later package more intuitive and easier to debug. Therefore, although our package `treedata.table` only extends the functionality of `data.table`s into the phylogenetic comparative workflow, this largely unexplored framework in the field will enable users to take advantage the speed and syntax of that is inherent to `data.table`.

### Methods

We used the `microbenchmark` (*Mersmann, 2019*) function under default parameters to compare the performance of functions in `treedata.table` to other packages (Appendix S3). First, we compared the performance in the initial tree/data matching step between `treedata.table` and `treeplyr` (`treedata.table::as.treedata.table()` and `treeplyr::make.treedata()`). We simulated trees with 10, 40, and 100 tips using rtree function in ape (*Paradis & Schliep, 2018*). Additionally, we generated random character matrices (50 discrete and 50 continuous traits) matching 90% of tips in the tree. Second, we compared the performance of data operations in `treedata.table` relative to `data.table` (*Dowle & Srinivasan, 2019*), base (*R Core Team, 2013*), `treeplyr` (*Uyeda & Harmon, 2020*), and dplyr (*Wickham et al., 2019a*; *Wickham et al., 2019b*). This time, we simulated trees with 1000, 10000, and 500000 tips using the `rtree` function in ape. Again, we generated random character matrices (50 discrete and 50 continuous traits) matching 90% of tips in simulated trees. We compared the performance of `treedata.table::[`, `data.table::[` `treeplyr::%>%`, `dplyr::%>%`, and the equivalent functions in base when (1) subsampling the full dataset for rows matching a single level in one discrete character,

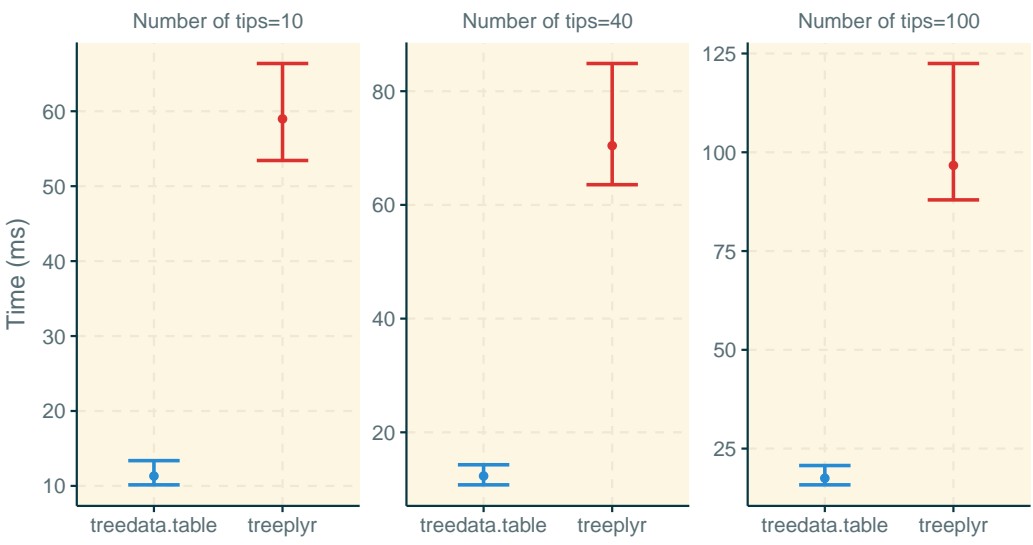

**Figure 2 Results for the `treedata.table` microbenchmark during tree/data matching steps.** Estimates of the timing during the tree/data matching steps under `treedata.table` are shown in relation to treeplyr. We show median and lower/upper quartiles times for the performance of each package.

and (2) estimating the sum and mean of two continuous traits based on the groups of a second discrete character. In `data.table` syntax for this process would be:

```
td$dat[Disc1 == "A", .(sum(Cont2), mean(Cont3)), by = Disc10]
```

## Results

`treedata.table` was >400% faster than `treeplyr` during the initial data/tree matching step (Fig. 2). For instance, combining a dataset with 10 tips to a character matrix of 40 traits (10% of unmatched tips), `as.treedata.table` takes an average of 12.314 ms (range = 8.100–27.479 ms) relative to the 64.198 ms that were needed in `treeplyr` (range = 48.407–166.328 ms). Differences in the performance between these two functions also scale with the number of taxa. Next, we examined the performance of data operations in `treedata.table` relative to `data.table,` base, treeplyr, and dplyr (Fig. 3). We found that the simultaneous processing of phylogenetic trees in `treedata.table`'s compromised the speed of our package by 90% relative to `data.table`. However, data manipulation in `treedata.table` (which simultaneously processes phylogenies) is still significantly faster than in other commonly used packages for data manipulation only, such as base (>35%), treeplyr (>60%), and dplyr (>90%). The higher speed performance of `treedata.table` relative to other functions also increases with the size of the dataset.

## CURRENT LIMITATIONS OF `treedata.table`

The current release of `treedata.table` can handle `phylo` and `multiPhylo` objects. A single character matrix is shared across all the trees in the `treedata.table` object. Additionally, all the trees and the only character matrix in the same `treedata.table` object are forced to have the same tip-level sampling. We acknowledge that partial tree/data matching is

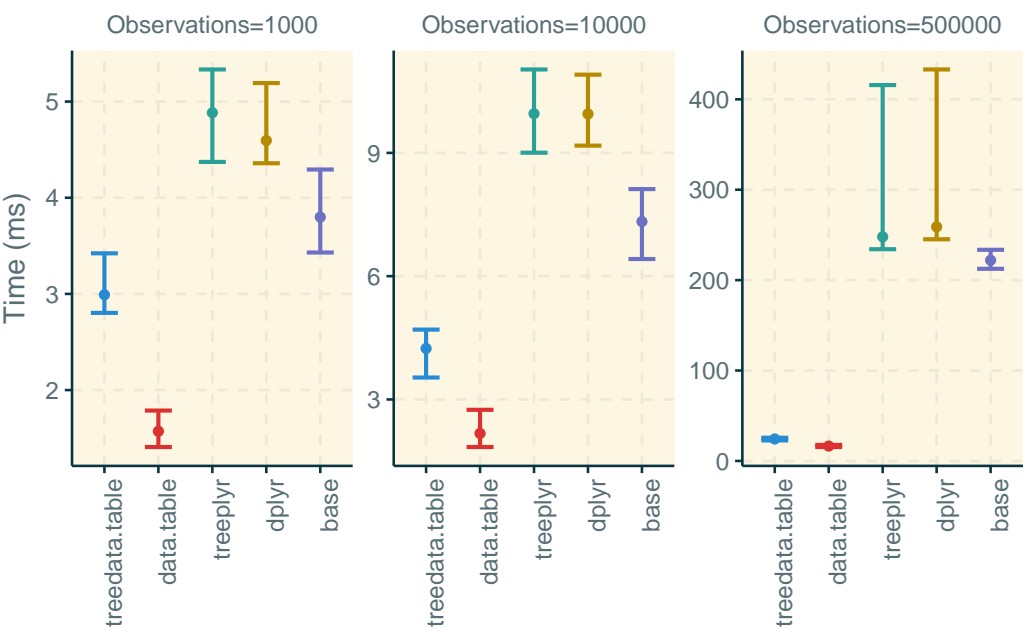

**Figure 3  Results for the `treedata.table` microbenchmark during data manipulation.** We compare the performance of `treedata.table` against `data.table`, base, treeplyr, and dplyr. We show median and lower/upper quartiles times for the performance of each package.

desirable in some situations. For instance, users may be interested in performing analyses on trees that, despite having different tip-level sampling, partially overlap with a common character matrix. Similarly, users may be interested in using multiple character matrices instead of only one. Future releases of the `treedata.table` package will focus on relaxing some restrictions on the tree/data matching.

## CONCLUSIONS

Here we describe the first R package that extends the functionality and syntax of `data.table` for performing operations in phylogenetic comparative datasets. We also note that `treedata.table` significantly improves the speed of the analytical workflow when compared to alternative methods for manipulating phylogenetic comparative data. `treedata.table` is expected to increase code reproducibility while simplifying the complexity of scripts. Finally, data manipulation in `treedata.table`, which is significantly faster than in other commonly used packages, will allow researchers to quickly perform data manipulation on large datasets without requiring outstanding computational resources.

## ACKNOWLEDGEMENTS

We thank Luke Harmon for his contributions that laid the groundwork for the current package. The authors thank Hugo Gruson, Kari Norman, Julia Gustavsen, Luna L. Sanchéz, and Guangchuang Yu for helpful comments during review in rOpenSci and PeerJ. Heidi E. Steiner revised an early version of the manuscript and assisted with logo design.

### Funding

This package was partially developed during the "Nantucket phylogeny developeR workshop", organized by Liam J. Revell (NSF DBI-1759940). April Wright was supported by an Institutional Development Award (IDeA) from the National Institute of General Medical Sciences of the National Institutes of Health under grant number P2O GM103424-18. Josef Uyeda was funded on NSF DEB-1208912 to Luke Harmon in creating `treeplyr`, which served as a precursor for this project and code. The funders had no role in study design, data collection and analysis, decision to publish, or preparation of the manuscript.

### Grant Disclosures

The following grant information was disclosed by the authors:
Nantucket Phylogeny DevelopeR Workshop: NSF DBI-1759940.
National Institute of General Medical Sciences of the National Institutes of Health: P2O GM103424-18.
NSF: DEB-1208912.

### Competing Interests

The authors declare there are no competing interests.

### Author Contributions

- Cristian Román Palacios, April Wright and Josef Uyeda conceived and designed the experiments, performed the experiments, analyzed the data, prepared figures and/or tables, authored or reviewed drafts of the paper, and approved the final draft.

### Data Availability

The `treedata.table` is available at https://ropensci.github.io/treedata.table/index.html.

### Supplemental Information

Supplemental information for this article can be found online at http://dx.doi.org/10.7717/peerj.12450#supplemental-information.

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
