# Peer review of "treedata.table: a wrapper for data.table that enables fast manipulation of large phylogenetic trees matched to data"

_PeerJ, doi:10.7717/peerj.12450_

## Round 0.1 · original submission · Major Revisions

Dear Dr. Palacios and colleagues:

Thanks for submitting your manuscript to PeerJ. I have now received two independent reviews of your work, and as you will see, the reviewers raised some concerns about the research. Despite this, these reviewers are optimistic about your work and the potential impact it will have on research studying the manipulation of large-scale sequence datasets for phylogeny estimation. Thus, I encourage you to revise your manuscript, accordingly, taking into account all of the concerns raised by both reviewers.

While the concerns of the reviewers are relatively minor, this is a major revision to ensure that the original reviewers have a chance to evaluate your responses to their concerns. There are many suggestions, which I am sure will greatly improve your manuscript once addressed.

Please ensure to make all aspects of your research reproducible; specifically, please share the code and results of the package benchmarking, as well as the raw data and code of the model fitting of temporal trend in tree size over time presented in Figure 1.
If possible, share the code used to generate all figures. Please provide a summary of comparisons demonstrating the superiority of treedata.table.

Therefore, I am recommending that you revise your manuscript, accordingly, taking into account all of the issues raised by the reviewers.

Good luck with your revision,

-joe

·

Basic reporting

• I commend the authors for presenting a professional structure of the manuscript, figures and tables.
• Similarly, a clear and unambiguous, professional English has been used throughout the manuscript and
package website.
• Moreover, supplementary material has been very well prepared, and has been made available from
the journal and from the software package website at https://ropensci.github.io/treedata.table/.
Supplemental files are written in English and Spanish. This is a very good step towards inclusivity. It
will also facilitate broader adoption of the package.
• Data to run all the example code has been shared as part of the software package. However, data from
the microbenchmarking of the package that were used to generate Figure 2 and 3, and raw data used
to generate Figure 1 have not been shared. For the sake of reproducibility, please share a link to the
repositories containing these data.
• The paper is self-contained, and results presented are relevant to the research goal: developing an R
package that facilitates fast and reproducible comparative phylogenetics analysis in large trees. I think
specifying in the title that the package is designed to deal with large trees (“fast manipulation of large
trees” instead of “trees” in general), will help readers immediately get a sense of the advantage of
learning and eventually using the package.
• The biological background of the package, as well as references supporting its importance are adequate.
The biological introduction is clear and relevant for the target audience. I would try to be more
specific in L42, stating that you are talking about the workflow for comparative phylogenetics and not
phylogenetics in general.
• The computational background can be improved by mentioning other packages that also deal with data
tables, maybe talk about the R tidyverse principles and argumenting why that structure is or is not
adopted for this package. I would consider making the section “A short description of data.table” part
of the introduction. In my opinion, this would round up the intro nicely.

Experimental design

• The authors present a research goal that is well defined, relevant and meaningful for the fields
of evolutionary biology and ecology. Authors successfully identify an analysis gap in these fields:
automatically creating comparative phylogenetics datasets with large trees in a fast and reproducible
way. The authors clearly state and demonstrate throughout the manuscript how the package they
present fills this gap.
• In this sense, the authors have developed original primary research that is within the Aims and Scope
of the journal.
• To achieve their goal, the authors develop a package written in the R language, that expands on the
fastest R package available for manipulation of data tables, the data.table package, to allow tree
manipulation. The authors appropriately justify the use of the R language, and present a rigorous
research of the available R packages for data table manipulation, and appropriately chose the one that
would give the fastest results for large tree manipulation. This should be made more explicit in the
introduction. For example in L52, they should mention the dplyr package and the data.frame workflow
from base.
• The authors described the package functions and usage very clearly and thoroughly. To test the speed
of the package they use the microbenchmark workflow, which is a current good standard for R package
benchmarking. However, the exact code used was not shared. This as well as the R data objects that
resulted from the microbenchmark runs should be shared as raw data.
• Statistical analysis used to analyze the temporal trend in phylogenetic tree size over time is adequate.
Code and raw data should also be shared.

Validity of the findings

• Following PeerJ recommendations, I am not leading the review with my perception on how impactful
and novel the research is. Nevertheless, I do believe this package will have a positive impact on the
field. I was able to successfully install the package from CRAN, and to run all example code in the ms
and supplementary data, and to reproduce the reported results exactly.
• Whenever possible, underlying data and code used to generate all figures should be provided.
• Conclusions are clearly stated, linked to original research question and limited to supporting results,
demonstrating that the treedata.table package effectively extends the functionality and syntax of
data.table for evolutionary research. However, the benchmark results they present seem to contradict
their conclusion that “treedata.table extends the speed of data.table”, in L230-231. Unless they mean
in comparison to treeplyr. But from the text I was unable to infer if treeplyr is also an extension of
data.table. If so, then treedata.table is not the first package to do this? This should be stated more
clearly.

Additional comments

To the manuscript:
• Adding a link to the website of the package (in the intro and abstract) would help readers know right
away that the package has been already published and is available for download. It would also prompt
readers to go explore it right away.
• I would highly recommend formatting function names with backticks (``) or quoting them. It would
help with readibility of the paper, especially for functions such as [ or operators such as $.
• L24: increasing -> increased
• L44-45: This language -> The R language
• L53-54: https://h2oai.github.io/db-benchmark, this should be a proper citation of the website with
date accessed. See an example of citation guidelines for websites here.
• L101: matrixes -> matrices
• L120-124: As I stated in general comments section, I think the output of these functions (especially
summary) should be described more, and/or refer readers to Table 1.
• L127: I suggest being more descriptive. Instead of saying “using the following line”, explicitly state the
function that is going to be used, e.g., “using the $ operator and the[ function, as follows:”
• L149: “furthermore, can create” -> “furthermore, we can create”
• L150: anoles. -> anoles:
• L171: Add a link to the vignette and specify which one of the 5 vignettes available.
• L211: “relative to” is repeated twice.
• L214: a comma is missing after “manipulation only”. Also, because you only tested speed of performance
among packages, “higher performance” should be “higher speed performance”.
• Figure 1:
– Legend: When using the rotl package, the Open Tree of Life project should also be cited, as
mentioned here https://cran.r-project.org/web/packages/rotl/citation.html, as Open Tree of
Life, B. Redelings, L.L. Sanchez Reyes, K.A. Cranston, J. Allman, M.T. Holder, & E.J. McTavish.
(2019). Open Tree of Life Synthetic Tree (Version 12.3). Zenodo. doi: 10.5281/zenodo.3937741
– When using log scale for a variable that is better understood in a non-logarithmic form, it is
best to replace the log numbers for their non-log equivalents. In this case the y axis showing the
number of tips can be modified to make the figure more readable, by replacing 1.6 for ~5, 3.2 by
~25, . . . , 8 by ~3k.
• Table 1: In the description column, I would remove “Function to . . . ”, it would save space and make it
less repetitive. You can start right away with the action that the function is performing. I would also
homogenize the verbs to present tense instead of using “ing” verbs sometimes.
• Figures in treedata.tables website preprint vignette appear broken, here: https://ropensci.github.io
/treedata.table/articles/E_Preprint.html.

To the README.md:
• add library(treedata.table) after installation instructions.
• in Line 54, explain that you are talking about function tdt and extractVector before using them in
line 57. It seems like you are only talking about functions from the geiger package, so it was not fully
intuitive for me to figure out the example in L57.
• add a license and citation
• section tl;dr seems out of place? It seems to me like it would go better as part of the first paragraph of
the README intro, maybe?
• L52: intutive -> intuitive
• L68: tipcs -> tips
• throughout the text: taxa names -> taxon names

To the AA_treedata.table_intro_english.Rmd file:
• L37 and L40: Some users might want to use your package without knowing about data.table. Try to
explain the main difference between data.table and data.frame. Maybe printing the class of the object
before and after would help readers visualize how the data object changes in structure.
• I like the named vectors!
• L78: Explain partial match and non-standard evaluation. Why are they relevant to be implemented in
your package? Linking partial matching to the corresponding vignette would be good.
• L80: consider a more straightforward wording for “all the tips not in the resultant data.table ”, to
maybe “all the tips absent from the resultant data.table ”
• Consider breaking down the Manipulating data section in subtopics. I would suggest at least making a
subtitle for Coindexing, so you can highlight this powerful functionality of your package.
• L101: again consider that readers might not be familiar with data.table, so a comparison with it is not
direct for all (most) users. It would make adoption of the package easier if the descriptions were a bit
more general, sort of speak.
• L107: the same goes for the concept of a “tidy approach”. Old school R users might not be familiar
with it, and new users might not have learned it yet. Try to think of a way to describe it in the simplest,
most general words.
• L129: Not sure I have it right, but maybe replace “out” for “the”?
To the file AB_treedata.table_intro_spanish.Rmd:
• It would be good to make the same editions I suggested for the English version of this vignette, i.e.,
making more subtitles, reducing the size of printed tables, elaborating more on “evaluación no estándar”,
“correspondencia o emparejado parcial”, “sintaxis del tidyverse”. And trying to explain things from
the point of view of someone that has little experience using the tidyverse, data.table and even R,
elaborating more on the comparisons with data.table.
• I am a native Spanish speaker, so I went ahead and checked typos and I made some suggestions to
change some of the wording to make it clearer. Feel free to disregard them or double check with other
Spanish speakers!
– L22:
* Anolis should be in italics
* arboles -> árboles
* están en formato -> estar en formato
– L18, 20 and 50: “Manipulando” should be “Manejando”. “Manipular” has a somewhat negative
connotation in Spanish, meaning that the data have been altered to change the results, or to
mislead.
– L65:
* “ser logrado” -> “ser obtenido”
* “El producto de extractVector como los brackets dobles es un vector con nombres” -> “Al
igual que con los brackets dobles, el resultado de la función extractVector es un vector con
nombres.”
– L78:
* I would remove the “Sin embargo”, it is not needed.
* match parcial -> correspondencia parcial
* múltiples columna -> múltiples columnas
* no standard -> no estándar
– L102:
* “pueden ser también operadas usando sintaxis en data.table” -> “pueden ser operadas
usando la misma sintaxis de data.table”
* arboles -> árboles
* simultanea -> simultánea
* y resumir -> y reducir
* numero -> número
– L108:
* “También podemos usar treedata.table para correr funciones en nuestros datos.” ->
“LA”treedata.table permite aplicar funciones directamente en nuestros datos de interés"
– L 116: “Los tips en el árbol también pueden ser removidos” -> “Las puntas del árbol también
pueden ser removidas”
– L135:
* ultimo -> último
* arbol -> árbol
* extraidos -> extraídos

To the B_multiphylo_treedata.table.Rmd file:
• You do a great job describing the output for the multiphylo objects. By reading the vignette, it was
not clear to me what the dataset output would be using a multiphylo object. Knowing that all trees
must have the same tip labels, I think I was able to infer what it would be, but it is always good to
have confirmation from the authors. It would be good to clarify in the text that:
– there is only one output dataset and not one for each phy in the multiphylo object
– the output dataset contains only the overlapping taxa between the multiphylo objects and the
input dataset
• L33: Use of “Nevertheless” here was confusing for me. As I understand it, “nevertheless” is an adverb
used to contrast a first point with a second one. I do not easily see that the two points contrast with
each other (as they both indicate a restriction?): “all trees must have the same tip labels. Nevertheless,
both the provided multiPhylo and data.frame should partially overlap.” Maybe it is just a matter of
replacing “should” by “can”. If the trees and datasets have to partially overlap, then I would replace
“nevertheless” by “also” or “and”, just to make it straightforward for readers to understand your point.

To the C_PartialMatching.Rmd file:
• L31, 47, and 56: ahi should be ahli.

To the D_AdditionalFunctions_treedata.table.Rmd file:
• L18: character -> characters
To the vignettes in general:
• when dealing with large tables, consider doing head and tail instead, or printing an interactive table,
otherwise the text and explanations get a bit lost among the long printed outputs of the example code.

To the code in general:
• I really like the messages you print to screen. They are short, sweet, and helpful.
• Try to always keep messages affirmative, as far as possible. Somehow negative messages always feel to
me like an error and leave me feeling that I did not run the function correctly. For example, instead of
No tips were dropped from the original tree/dataset you can print All tips from original
tree/dataset were preserved. This will be kinder for your users.
• When trying the examples with the modified tip ahli to NAA, I noticed that the output message is
Tip labels detected in column: X
Phylo object detected
1 tip(s) dropped from the original tree
1 tip(s) dropped from the original dataset"
In the last line, instead of "tip(s)" should be "line(s)" or "row(s)", or maybe change "tip(s)" to
"taxon(taxa)" in both cases? Not sure about that, maybe the first suggestion is better. I leave that to you
:)

·

Basic reporting

The package is quite simple and the manuscript is easy to follow. However, the review of relative works is not sufficient. The authors claimed in the manuscript that only treeplyr can combine tree with data. This is not true, there are several packages support this feature, such as phylobase, treeio and tidytree. The tidytree package is the fundamental package that the famous ggtree relies on. It can convert tree with data (parsed by treeio) into tidy data frame and supports dplyr verbs to manipulate tree with data. More packages should be included in the "Alternatives to treedata.table" session.

Experimental design

I have no doubt that treedata.table will have higher performance compare to other packages. However, as a user, I would like to see more comparisons on functionalities. Which packages can support the operation I needed is something more important. A table for detailed comparison is wanted.

Validity of the findings

In addition to speed comparison, more results on functionality comparison should also be included.

Additional comments

The limitation of this package is quite obvious that it only supports matching taxa with labels. This will restrict its application on internal nodes and on trees with identical tip labels.

---

## Round 0.2 · accepted · Accept

Dear Dr. Palacios and colleagues:

Thanks for revising your manuscript based on the concerns raised by the reviewers. I now believe that your manuscript is suitable for publication. Congratulations! I look forward to seeing this work in print, and I anticipate it being an important resource for groups studying the manipulation of large-scale sequence datasets for phylogeny estimation. Thanks again for choosing PeerJ to publish such important work.

Best,

-joe